# Equine-Assisted Interventions: Cross Perspectives of Beneficiaries and Their Caregivers from a Qualitative Perspective

**DOI:** 10.3390/geriatrics10060145

**Published:** 2025-11-06

**Authors:** Léa Badin, Elina Van Dendaele, Nathalie Bailly

**Affiliations:** Department of Psychology, University of Tours, UR 2114, 37000 Tours, France; lea.badin@univ-tours.fr (L.B.);

**Keywords:** Alzheimer’s disease, equine-assisted interventions, family caregiver, nursing homes, professional caregiver

## Abstract

**Background**: Although equine-assisted interventions (EAI) are gaining growing attention, their scientific evaluation among individuals with Alzheimer’s disease (AD) living in nursing homes remains limited. This study aimed to explore the lived experiences of an EAI program from the perspectives of the participants living with AD as well as their families and professional caregivers. **Methods**: Thirty non-directive interviews were conducted between June and July 2024 across several nursing homes in the Centre-Val de Loire region (France). The interviews were recorded, transcribed, and analyzed using thematic analysis. **Results**: Four main themes emerged from the analysis: (1) the experience with the horse, reflecting a unique relationship with the animal, the activities carried out, and perceived personality traits; (2) the environment of EAI sessions, offering a break from daily routines, encouraging contact with nature, and taking place in a setting specific to this type of intervention; (3) the implementation of the program within the institutional context, highlighting logistical aspects, environmental factors, and the adherence; (4) the effects of the intervention, including enhanced social interactions, memory stimulation, emotional engagement, and behavioral benefits. **Conclusions**: These findings provide insight into the multiple dimensions involved in an EAI program. By giving voice to both participants and their caregivers, this study emphasizes the value of qualitative approaches in deeply understanding the meaning and impact of these non-pharmacological interventions.

## 1. Introduction

Today, Alzheimer’s disease (AD), a neurodegenerative condition affecting multiple cognitive domains represents a major global public health challenge. It significantly impacts the quality of life of both the individuals concerned and their relatives [1]. As the disease progresses with advancing age, more than half of the population aged around 60 (i.e., 150 million people by 2050) will be affected by AD [2]. To date, numerous therapeutic treatments have been tested, but no curative results have been demonstrated [3]. As a result, a shift in therapeutic strategy has been adopted, focusing on the use of non-pharmacological interventions to maintain and/or improve the quality of life of individuals with AD [1]. Among these, Animal-Assisted Interventions (AAI) have shown promising results in terms of cognitive functioning and mood [4]. They also appear to enhance physical functioning, such as improvements in movement and balance [5], and social functioning, by fostering communication and social inclusion [6]. Today, most AAI involving older adults focus on interactions with dogs [7].

In this study, we have focused on Equine-Assisted Interventions (EAI). These interventions are defined as involving a horse as a mediator in the patient–professional relationship, with the aim of enhancing the beneficiary’s health and well-being. EAIs encompass a wide range of activities, such as grooming, building a connection with the horse, mobility exercises, walking, and observing horses in their natural environment. EAI programs vary in frequency and duration depending on the intended goals: recreational (Equine-Assisted Activities, EAA) or therapeutic (Equine-Assisted Therapies, EAT) [8,9] and have demonstrated promising effects. Indeed, most available scientific data on the effects of EAI concerns young populations with autism spectrum disorders. These studies show that EAI facilitates social functioning and relationships, increases engagement, reduces so-called “challenging” behaviors, and, from a cognitive perspective, reduces reaction time in problem-solving situations [10,11,12,13].

Some studies also highlight benefits among older adults, particularly with regard to physical health, showing improvements in balance, gait, and muscle strength [8,14,15,16] as well as psychosocial health, with improved quality of life [17,18].

In a review, Sebalj et al. (2024) [19] explored the use of EAI among older adults with AD, identifying six relevant studies encompassing various methodologies. The findings revealed positive outcomes across multiple domains, including social engagement [17,20,21], emotional well-being [17,20,21,22], behavioral aspects [22], and physical health [16]. A recent study was also conducted among older adults living with Alzheimer’s disease, in which a 12 session EAI program was implemented in an equestrian center. The results showed improvements in neuropsychiatric symptoms, quality of life, and a slight reduction in depressive symptoms [23]. Together, these results underscore the potential of EAIs to enhance the health and quality of life of this population.

Nevertheless, most studies on the benefits of EAI rely on quantitative methods to interpret the results, and to our knowledge, few have used a qualitative approach to explore these effects. As qualitative research in this field remains extremely scarce, if not entirely absent, it is essential to explore how older adults with AD perceive, interpret, and make sense of their participation in EAI, particularly regarding their subjective and experiential dimensions. Likewise, the perspectives and experiences of families and professional caregivers regarding these sessions are poorly documented [19].

The aim of this study was therefore to explore the subjective experiences of individuals with AD who have participated in EAI sessions, as well as those of their families and professional caregivers, to better understand their perceived perspectives regarding these interventions.

## 2. Materials and Methods

### 2.1. Participants

In this study, 30 individuals (older adults: *n* = 10; family caregivers: *n* = 10; professional caregivers: *n* = 10) were included, of whom the older adults had previously participated in an earlier study examining the effects of equine-assisted interventions (EAI) on psychological health [23]. In that previous study, older adults living with Alzheimer’s disease (MMSE scores: 10 to 20) and residing in nursing homes participated in a 12-week EAI program consisting of one session per week. The sessions were conducted outdoors at an equestrian center and were structured around three main phases: grooming, exercises in the riding area, and reward.

Recruitment for the present study was carried out in collaboration with psychologists from partner institutions and a member of the research team, based on predefined inclusion and exclusion criteria. The psychologists contacted family and professional caregivers, and participation was voluntary.

For older adults living with AD, the inclusion criteria were as follows: (1) participation in the previous study [23] (with main inclusion criteria being as follows: age ≥ 60 years, a diagnosis of Alzheimer’s disease, MMSE scores between 10 and 20, ability to walk without a wheelchair, and no fear or known allergy to horses.); and (2) completion of at least 10 out of 12 EAI sessions.

Family caregivers were eligible if they (1) were adults (≥18 years); and (2) had provided care at least once per week during the past four consecutive months.

Professional caregivers were included if they (1) were adults; (2) had been employed since February 2024; and (3) worked directly with EAI recipients. Common exclusion criteria across all participant groups included the following: (1) major sensory impairments that could hinder the interview process; and (2) significant difficulties in expressing themselves in French. An additional exclusion criterion applied to professional caregivers occupying non-fixed positions (i.e., rotating staff, meaning professionals who regularly move between different units within the nursing home and therefore cannot provide daily care to the same older adults). The final sample consisted of 10 older adults with AD, 10 family caregivers, and 10 professional caregivers.

### 2.2. Procedure

Each participant was interviewed individually and in person. Residents were met in their living space, while both family and professional caregivers were interviewed in a designated room within the care facility. Prior to each interview, participants were asked whether they experienced any conditions that might hinder their ability to participate (e.g., hearing or speech impairments, significant fatigue) and whether they felt able to proceed. They were informed of the voluntary and anonymous nature of the study, their right to withdraw their data at any time, and the audio recording of the interview. Written informed consent was obtained from all participants. Basic sociodemographic information (age, gender, and occupation) was collected prior to the interview, which was conducted entirely in French. Non-directed interviews were conducted to explore in depth the perceptions, opinions, and experiences of participants regarding equine-assisted interventions (EAI). This open approach, without a predefined interview guide, aimed to elicit rich and spontaneous data by allowing participants to express themselves freely. A single introductory question was used to initiate the discussion: “You or your relative, Mrs. X or Mr. Z, participated in the equine-assisted sessions. What can you tell me about that experience?” The interviewer took some handwritten notes during the interview. Concerning the duration, the interviews were not predetermined. They could be interrupted at the participant’s initiative or when the content became redundant, indicating data saturation.

The interviews were conducted between June and July 2024. For older adults living with AD, the interviews took place within a timeframe of 1 to 7 days following their last EAI session, to capture accounts that were as close as possible to the lived experience.

### 2.3. Analytical Approach

The qualitative methodology was developed and reported in accordance with the COREQ (Consolidated Criteria for Reporting Qualitative Research) checklist [24] (See Appendix A), which ensures transparency and rigor in qualitative research reporting. In accordance with this approach, an inductive thematic analysis was conducted to explore the experiences of beneficiaries of equine-assisted interventions (EAI), as well as those of their family and professional caregivers. The number of interviews (*n* = 10 per group) was determined a priori for both practical and comparative reasons, as redundancy in themes was observed among participants. Furthermore, the literature generally indicates that data saturation is often reached between 6 and 12 interviews [25]. In the present study, saturation was assessed using the frequency counting method, which consists of counting the new codes appearing in successive transcripts until no new codes are identified [25,26,27].

The analysis followed the six-phase approach outlined by Braun and Clarke (2006) [28]. We emphasize the importance of the reflexive process in the context of a qualitative study [29]. Indeed, in this type of research, the researchers themselves serve as the primary instrument of analysis, which requires a continuous process of reflexivity. In this study, the interviewer was a female psychologist specialized in aging and a PhD student conducting research on Alzheimer’s disease and equine-assisted interventions. Authors (LB, EVD, NB) were involved in the six steps in the following manner: (1) The interviews were conducted and transcribed by the first author (LB), who then re-read each transcript several times to gain deep familiarity with the data. This stage included note-taking to capture emerging ideas and impressions. (2) Initial manual coding was performed by the same author to identify relevant units of meaning (LB). The generated codes were then presented and discussed collectively to validate, refine, or complete them (LB, EVD, NB). (3) Codes were grouped into potential themes through in-depth discussions among the research team. A collaborative table was used to visualize the relationships between codes and emerging themes (LB, EVD, NB). (4) Themes were reviewed, refined, renamed, or reorganized into sub-themes when necessary (LB, EVD, NB). An inter-rater coding press was implemented: LB, EVD, and NB each coded two interviews, and discrepancies were resolved through discussion. Inter-rater agreement was assessed using Fleiss’ kappa coefficient. The overall result (κ = 0.608) indicated a good level of agreement among the three coders. (5) Each theme was clearly defined and named, highlighting its contribution to understanding the research question and the potential links between themes (LB, EVD, NB). (6) The final analysis was collaboratively written, supported by representative excerpts, and discussed in relation to the study’s objectives and existing literature (LB, EVD, NB). In parallel, descriptive statistical analyses were performed to characterize participants’ sociodemographic data. In addition, descriptive statistics (percentages) were also conducted to present the themes addressed by our three populations.

For this analytical phase, the software used included the Microsoft Word (2024) transcriber for transcription, Microsoft Excel (2024) for the development of the codebook, and the statistical software JASP (0.17.2.1.) for conducting descriptive analyses (percentages) and calculating Fleiss’ kappa coefficient.

### 2.4. Ethics

This study received approval from the Ethics Committee for Research Involving Human Subjects at the Universities of Tours and Poitiers (ref: CER-TP 2024-02-01). In accordance with current ethical standards, an information sheet and an informed consent form were provided to each participant. Participants were informed prior to each interview that it would be recorded, and recordings were only made with their consent.

## 3. Results

### 3.1. Characteristics of Participants

In this study, 30 interviews were conducted, including 10 older adults with AD, 10 family caregivers, and 10 professional caregivers. Sociodemographic characteristics are presented in Table 1. The sample included 20 women and 10 men. The mean age of the older adults was 85.7 (SD = 4.83), the mean age of the family caregivers was 65.4 (SD = 9.17), and the mean age of the professional caregivers was 37.8 (SD = 12.1).

### 3.2. Main Themes

From these 30 interviews, four main themes were identified: (1) perceived effects of EAI, (2) the implementation of EAI, (3) the environment of EAI, and (4) experience with the horse. Table 2 presents these themes and their associated subthemes, along with the number and percentage of occurrences across the 30 interviews. The table also highlights differences observed between the three participant groups: older adults with AD (OA), family caregivers (FC), and professional caregivers (PC).

#### 3.2.1. Perspectives from the Three Populations

A comparison of the discourse across the three participant groups reveals distinct thematic priorities. Among older adults, the theme of the effects of EAI is predominant, mentioned by all participants in this group (*n* = 10; 100%). Family caregivers addressed three of the main themes equally, each reported by 100% of them: the perceived effects of the EAI, the conditions surrounding its implementation, and the environment in which it takes place. Finally, professional caregivers placed greater emphasis on the environment and the effects of the EAI (*n* = 9; 90%).

#### 3.2.2. Theme 1: Effects Induced by the EAI Program

The participants in this study brought out a fourth theme: the effects induced by the EAI program (*n* = 30). This theme includes several subthemes: (a) Social interactions (*n* = 24), (b) Memory stimulation (*n* = 21), (c) Emergence of emotions (*n* = 27), and (d) Behaviors induced by the EAI program (*n* = 15).

The first sub-theme related to the effects of the EAI program concerns the emergence of social interactions. Older adults (*n* = 7) said they appreciated being in a small group: “Well, there were friends too. They were with me… We were 3 or 4, yeah, and that was nice.” (Older adult 5). All interviewed family caregivers (*n* = 10) mentioned renewed exchanges with their loved one, with some even describing the return of a sense of closeness:

“On the other hand, I really enjoyed those equine therapy sessions because I found back some of the funny sides of my mother that she didn’t have anymore […] I laughed a lot with her, we had shared glances, I had that closeness with her again that I haven’t had in a very long time… And I found my mom’s sense of humor again.” (Family caregiver 4).

All professional caregivers (*n* = 10) also mentioned positive interactions with the residents and emphasized their appreciation of the collective involvement of staff during EAI sessions: “The fact that the managers could also interact with us, it kind of breaks down the hierarchy between management and employees.” (Professional caregiver 9).

The second sub-theme concerns memory stimulation. Older adults (*n* = 9) remembered specific elements from the sessions: the names of equipment, activities, or even the ponies’ names: “I only remembered the French name, it’s Bleu! Oh no, it’s Blue.” (Older adult 5). Family caregivers (*n* = 9) spoke about memories their loved ones associated with the sessions, despite the progression of the disease:

“Well, I got the impression that during Scrabble, she wrote little words, she talked about the longe line… Well, that came out pretty quickly after the third or fourth equine therapy session. I got to hear about the longe.” (Family caregiver 4)

Professional caregivers (*n* = 3) reported that some older adults spontaneously shared family memories triggered by the sessions: “He tells us memories he had with his father, who was in the same profession as him. So yes, it brings back some small memories too.” (Professional caregiver 10).

The third sub-theme highlights the emergence of emotions. Older adults (*n* = 8) expressed the joy they felt during the sessions: “I really liked it a lot, because it’s my dream to see those little creatures [the horses].” (Older adult 8). Family caregivers (*n* = 10) also said they were emotionally moved by those moments:

“But for me, it’s true, I used to come on Monday mornings, and I admit I was happy. I felt, how can I say, so cheerful, because I knew he would feel good… I wasn’t worried. I had music on in the car, I was happy, you see, it was like that…” (Family caregiver 8).

Professional caregivers (*n* = 9) spoke about the emotions they observed in residents after the EAI sessions: “She will have gone away, and she wouldn’t have been like that at all, and she’ll come back and she’ll be… I’d say glowing, she’ll really have a smile and… I don’t know how to explain it.” (Professional caregiver 3).

The fourth sub-theme discusses behaviors induced by the EAI sessions. Older adults (*n* = 8) reported a calming effect (i.e., less agitation)**:** “It’s more peaceful, much better (than in the nursing home).” (Older adult 5). From the family caregivers (*n* = 2), a sense of calm was also mentioned, both for their loved one and for themselves: “I had the feeling that those equine therapy sessions calmed him down somehow […] They helped me get through the winter, so to speak. Well, not winter, but spring.” (Family caregiver 4). Finally, half of the interviewed professional caregivers (*n* = 5) also noted this calming effect generated by the sessions:

“There, in that context, we see them completely differently, with symptoms that are managed in the moment. For example, we have a resident who complains all day long about heartburn. But there, during those moments, she doesn’t mention it at all. She shifts her focus away from those anxieties she usually has during the day.” (Professional caregiver 10).

#### 3.2.3. Theme 2: Implementation of the EAI Program

The participants in this study brought out a third theme: the implementation of the EAI program (*n* = 25). This theme includes several subthemes: (a) Institutional logistics (*n* = 16), (b) Leaving the nursing home (*n* = 13), and (c) Adherence to the program (*n* = 23).

The first identified sub-theme concerns the institutional logistics related to the implementation of the EAI program. More specifically, a few older adults (*n* = 2) mentioned the regularity of the sessions: “Yes. Every Tuesday, yes.” (Older adult 5). The family caregivers (*n* = 7), for their part, emphasized the organizational difficulties inherent in setting up the intervention: “I also understand that it’s not easy to organize, that staff is needed, and that logistically you also need vehicles to transport them.” (Family caregiver 9). They also highlighted a lack of communication about the program from the institution to families: “We’re not really informed, actually. There may be a lack of communication about it.” (Family caregiver 8). On the professional caregivers’ side (*n* = 7), some noted they had more time to dedicate to the residents during outings:

“In the morning, we can’t necessarily take care of them. I mean, take the time to talk with them like we can outside. In the morning, we’re in the bathrooms. But when we go on outings, we take the time to talk with them, we get to know them better.” (Professional caregiver 8).

The second emerging sub-theme refers to leaving the nursing home, which influenced the implementation of the program and was mainly mentioned by caregivers. Only one older adult mentioned the transportation required to get to the equestrian center: “And then the lady who takes us in her… we can’t really say car because it’s a big truck. So I got used to it just fine.” (Older adult 10). Family caregivers (*n* = 5) and professional caregivers (*n* = 7) highlighted the stressful nature of leaving the nursing home for older adults:

“But he was a bit unsettled because he was outside, so the transport, all that—it was maybe a bit unsettling for him, that’s my point of view.” (Family caregiver 1)

“Because generally, it’s pretty common with residents in a secure unit—when they leave their comfort zone, when they go outside… and when they come back to the original place where they spend most of their time… well, it becomes quite anxiety-inducing for them… Even if it’s not an activity, it could just be a family visit, people from their circle? All it takes is for them to be taken to another floor or the ground floor of the nursing home.” (Professional caregiver 2)

The third sub-theme relates to adherence to the EAI program. Older adults (*n* = 7) expressed a desire to continue the sessions: “Well, I hope it lasts a long time, that we’ll keep doing it for a long time.” (Older adult 8). The family caregivers (*n* = 7) mentioned how familiar and pleasant the equestrian center was for their relatives, which may have fostered their engagement: “And he’s someone who spent a lot of time around animals and nature. Yeah, so it’s always been a very, very pleasant environment for him.” (Family caregiver 5). The professional caregivers (*n* = 9) observed strong motivation among participants, particularly in how they cared for the pony:

“You could really see that she was thrilled to be there, and I was able to attend one of the sessions too, and I saw the motivation, to get up, not to complain about leg pain, to be happy to walk next to the horse or go up to it. It was impressive.” (Professional caregiver 7).

#### 3.2.4. Theme 3: Environment Within the EAI

The participants in this study brought out a second theme: Environment within the EAI (*n* = 24). This theme includes several subthemes: (a) Break from everyday life (*n* = 20), (b) Contact with nature (*n* = 8), and (c) Setting of the activity (*n* = 7).

The first identified sub-theme focuses on the break from daily routine induced by the implementation of the equine-assisted intervention program within the facility. Although few older adults (*n* = 2) mentioned this point, the idea of an extraordinary place emerged:

“He had beautiful places that felt like a dream.” (Older adult 10). From the family caregivers (*n* = 8), many of them emphasized the importance of the activity taking place outside of the usual nursing home setting, perceived as a beneficial opening: “It was really something that was outside the nursing home setting […] so offering something that takes place outside, I thought it was really, really great.” (Family caregiver 9). All the professional caregivers (*n* = 10) also recognized this break from routine, not only for the residents but also for themselves: “That’s really it actually, it’s getting out of our work routine and spending, let’s say, a special moment with residents we support every day, or almost every day, during our workdays.” (Professional caregiver 10).

A second emerging sub-theme concerned contact with nature. Although mentioned by few older adults (*n* = 2), this connection was nevertheless described as pleasant: “No, it’s being in the countryside, being with the horses, all of that is wonderful.” (Older adult 5). The family caregivers (*n* = 2), for their part, highlighted the importance of maintaining this connection with nature for their loved one’s well-being: “I know it brings him happiness. He loves looking at the flowers, the birds. I think it’s important to still be able to offer him that opportunity to marvel, to give him little bubbles of joy actually.” (Family caregiver 5). Finally, the professional caregivers (*n* = 4) emphasized the sense of freedom that this connection with the natural environment provides: “The feeling of being free, of seeing nature.” (Professional caregiver 8).

A third sub-theme concerns the setting in which the activity took place. Although relatively rarely mentioned, certain comments highlighted the impact of the location on the residents. Older adults (*n* = 3) said they were especially impressed by the size of the facilities:

“Oh yes, wow yes. Well, it’s so big, the hall. It’s huge. I don’t know how long or wide it is, it’s wonderful.” (Older adult 10). The family caregivers (*n* = 3) also reported their loved one’s amazement or wonder at the space where the sessions took place: “Yes, what she did and above all the fact that it was a big space, I suppose it was the arena.” (Family caregiver 9). From the professional caregiver’s point of view (*n* = 1), the equestrian center was perceived as a pleasant place, offering a true breath of fresh air in their daily routine: “Well, that little breath of fresh air [the equestrian center], getting out, really.” (Professional caregiver 5).

#### 3.2.5. Theme 4: Experience with the Horse

The participants in this study brought out a first theme: the experience with the horse. This theme includes several subthemes: (a) the relationship with the horse (*n* = 20), (b) activities carried out with the horse (*n* = 16), and (c) the characteristics of the horse (*n* = 12).

Concerning older adults, they describe in the first sub-theme a rather spontaneous and close relationship with the horse that was assigned to them (*n* = 8): “He came to me every time I arrived, I’d go to him and say: ‘Hello Boutch, how are you today? Are you feeling good?’ and I’d give him a kiss on his muzzle.” (Older adult 8). Some also report a calming and sought-after physical contact: “Yes yes, he was very gentle because he came close to me and stayed like that. So, I wanted to pet him.” (Older adult 9). From the family caregiver’s perspective (*n* = 6), they observed meaningful and emotional interactions between their loved one and the horse: “What touched her the most was that during the last session, her horse came and rubbed up against her, kind of to say thank you, and that really moved her.” (Family caregiver 7). As for the professional caregivers (*n* = 6), they perceived the horse’s presence as reassuring for the older adults: “And the contact with the animal, being able to… it was a somewhat reassuring presence for her.” (Professional caregiver 4).

Regarding the second sub-theme, which focuses on the activities with the horse proposed in the EAI program, the older adults (*n* = 8) mentioned their active involvement in grooming and feeding the horses: “Yes, we brushed them. They gave us things to brush them with. And we fed them.” (Older adult 5). Family caregivers (*n* = 5) reported stories told by their loved ones about these activities: “She said she was happy, she had brushed him, cleaned his hooves.” (Family caregivers 10). The professional caregivers (*n* = 3) highlighted the relevance of the proposed activities, which they found well adapted to the capacities of the older adults: “They were balance courses. For example, holding the pony, they had to do balance exercises, memory games, and games with road signs. These weren’t activities that didn’t suit them, in reality.” (Professional caregiver 6).

A third sub-theme related to their horse’s characteristics also emerged. The older adults (*n* = 6) said they were impressed by its intelligence and grace: “Because it’s true that when you see those little creatures, they’re intelligent, they let themselves be guided […] it’s a very very graceful animal.” (Older adult 7). Family caregivers (*n* = 3) emphasized the horses’ sociable and docile temperament: “They were used to seeing people, they were very sociable… They were easy to handle.” (Family caregiver 3). As for the professional caregivers (*n* = 3), they mentioned the horse’s ability to perceive human emotions: “Because horses feel everything […] if the horse sees you’re stressed, angry or something else, I think it can affect the horse’s behavior.” (Professional caregiver 9).

## 4. Discussion

This study aimed to explore the lived experience of an EAI program, both from the perspective of the participants themselves, who are living with Alzheimer’s disease, and that of their families and professional caregivers. The findings suggest that, in this context, the concept of EAI is not limited to the interaction between humans and animals alone. The environment in which the intervention takes place, as well as the broader effects (i.e., social interactions, memory stimulation, emergence of emotion, behavior) induced by the program, appear to play an equally important role. The experience thus emerges as multidimensional, shaped by the relationship with the animal, the environmental context, the implementation of the program, and the benefits derived from it.

### 4.1. Effects Induced by the EAI Program

Concerning the effects induced by the EAI program, they appear to be multidimensional: increased social interactions, memory stimulation, enhanced emotional expression, and reduction in certain behaviors. The collected verbatim accounts reveal a range of positive emotions experienced during the sessions. These findings are consistent with the literature. A study conducted among older adults with neurocognitive disorders highlighted an improvement in emotional well-being, specifically a sense of pleasure associated with participation in EAI [17]. One notable effect of EAI is the increase in social interactions. This social dynamic built around a shared point of interest illustrates what the literature refers to as the “social catalyst” function played by the animal [30]. Our findings align with those of Fields et al. (2018) [17], who showed that implementing an EAI program fostered more conversation among older adults with neurocognitive disorders compared to other activities offered in nursing homes. The horse thus becomes a topic of interest and a vehicle for exchange. The calming and stimulating power of contact with animals among older adults with Alzheimer’s disease is well documented, especially through canine-assisted therapy. Richeson’s (2003) [31] controlled study showed that therapy involving a dog significantly reduced agitation and psychotropic medication use. Our results suggest that EAI could produce similar effects (i.e., calming effect). This hypothesis is supported by the findings of one study [22], which observed a reduction in behavioral disorders among older adults with dementia who participated in EAI. A plausible physiological mechanism for these effects could be the decrease in cortisol levels induced by the sessions [32]. A more unexpected but noteworthy aspect concerns the cognitive domain. Given the current literature, no significant improvement in memory functions was expected. Indeed, studies by Richeson (2003) and Araujo et al. (2018) did not find any impact of animal-assisted interventions (with dogs or horses) on the cognitive functions of people with AD [16,17,18,19,20,21,22,23,24,25,26,27,28,29,30,31]. However, in our study, participants spontaneously recalled specific elements related to the EAI program. Family caregivers also reported memories shared by their relatives. Professional caregivers observed the reemergence of distant family memories following the sessions. This more ecological form of cognitive stimulation was perceived as beneficial, even if it does not necessarily result in measurable improvement. It is consistent with the opinion of 29% of professionals surveyed in a recent study, who believe that EAI helps stimulate patients’ residual cognitive abilities [33]. Finally, it should be noted that all of the positive effects mentioned also appear to have an impact on family caregivers. In a study by Sheung-Tak Cheng et al. (2016), caregivers stated that seeing their loved one “doing relatively well” gave them a sense of well-being [34]. Similarly, in our study, relatives reported an emotional benefit linked to their parent’s participation in the EAI program.

### 4.2. Implementation of the EAI Program

The results of our qualitative study show that implementing an EAI program in nursing homes raises significant organizational and logistical challenges. This theme emerged prominently in the interviews. Among the main obstacles, the need for smooth coordination between stakeholders (management, caregivers, external facilitators, families) and careful planning seem essential to us. Caregivers reported a structural lack of time, while families pointed to insufficient communication. These findings echo the existing literature: a survey of 663 professionals in the social care sector identified “institutional burden” as the main barrier to implementing animal-assisted interventions [33]. The location of the sessions also posed challenges. Held outside the facility, the trips occasionally caused stress for some residents, especially upon returning to the nursing home. Behavioral issues were sometimes observed, likely linked to the environmental shift. These observations are consistent with studies [35] and recommendations from the French National Health Authority [36], which emphasize the destabilizing effects of disrupted routines for people with AD. Engagement with the program, from both residents and caregivers, proved to be a key success factor. It was often supported by perceived positive outcomes from a meaningful activity. These benefits also enhanced caregivers’ sense of usefulness and their quality of work life [37]. Finally, previous studies have shown that institutionalized older adults rarely engage in meaningful activities [38]. In this regard, the EAI program offers a novel form of engagement, where the older person becomes an active participant, even a caregiver to the animal, thus helping restore self-esteem and identity [39].

### 4.3. Environment Within the EAI

The emergence of the environment theme in participants’ accounts can be interpreted through the lens of biophilia theory. According to this hypothesis, human beings possess an innate inclination to seek contact with nature and other forms of life [40,41,42]. In our study, several participants (i.e., older adults living with AD, family caregivers, and professional caregivers) spontaneously mentioned being outdoors, in a non-medicalized setting. This theme appeared particularly salient among family caregivers, who seemed to experience a sense of satisfaction at seeing their family member leave the nursing home and reconnect with a natural environment, a fundamental need, according to Wilson (2017) [42], yet too often neglected in institutional care settings. This break from the institutional environment, often experienced as monotonous or confined, appears to provide a therapeutic benefit that extends beyond the scope of the EAI. Green Care Farms exemplify this dynamic [43,44,45]. These structures, seen as alternatives to traditional nursing homes, integrate nature, animals, and outdoor activities into the daily lives of residents. They offer a more stimulating environment, encouraging meaningful activities, such as gardening or caring for animals [43,44,45]. Moreover, a recent study highlights that this type of environment also benefits healthcare professionals: it is positively correlated with improved well-being and health [46].

### 4.4. Experience with the Horse

Our study shows that older adults with AD develop a spontaneous and special experience with the horse. This unique experience manifests through various forms of connection: physical contact, talking to the animal, and a particularly strong emotional attachment, especially as reported by family caregivers. Professional caregivers, on the other hand, highlight the horse’s reassuring presence, which contributes to a sense of safety among the older adults. These observations are consistent with findings in the literature, which indicate that individuals with neurocognitive disorders often express a strong interest in animals, facilitating their engagement in animal-assisted intervention (AAI) programs [47]. Moreover, individual interactions with the animal appear to foster the development of unique emotional bonds, while other studies describe the horse as providing a sense of protection [48,49,50,51,52,53]. These privileged relationships take place within a context of diverse and tailored activities, such as grooming, walking alongside the horse, or obstacle courses, as mentioned by the older adults themselves. This observation echoes broader AAI literature, which shows that a variety of structured activities (including petting, grooming, talking, walking, and games) can enhance cognitive stimulation and social interaction in people with dementia [54,55,56]. Professional caregivers emphasized the adaptation of tasks to everyone’s capacities, which fostered participants’ engagement. In this respect, Zhang et al. (2024) [47] report that individualized and tailored interventions significantly support older adults’ involvement in AAI programs. Finally, the overall experience is enriched by the inherent characteristics of the horse. Several older adults expressed a sense of wonder when interacting with the animal, a feeling that resonates with the symbolic and cultural value historically attributed to horses [57]. Professional caregivers also described the horse as being able to perceive and respond to human emotional states, a phenomenon supported by literature on emotional contagion between humans and horses [58,59]. This relational process aligns with the Integrative Model of Human–Animal Interaction (IMHAI) [60], which emphasizes shared emotional systems and interspecies co-regulation as key mechanisms underlying the benefits of animal assisted interventions.

Thus, the richness of the equine-assisted experience lies in the delicate interplay between emotional connection, adapted activities, and the horse’s relational and sensitive qualities.

### 4.5. Limits

This study presents several limitations. First, our sample does not reflect the entirety of the individuals who participated in the EAI program. Although the program involved 25 older adults, only 10 were interviewed for this research. Second, the issue of social desirability bias should be acknowledged. In qualitative research, participants may tend to present their experiences in ways that conform to perceived social norms. This bias can complicate the interpretation of results [61]. Furthermore, the interviewer was also involved in the data analysis, which contributed to a deeper understanding of the participants’ experiences. To ensure reflexivity and limit potential bias, a reflexive posture was maintained throughout the process, supported by continuous reference to raw transcripts and regular discussions with other members of the research team.

## 5. Conclusions

Although equine-assisted interventions (EAI) are increasingly recognized among non-pharmacological approaches, few studies have explored how they are experienced by people living with Alzheimer’s disease, their relatives, and care professionals in nursing homes. This study sheds light on the meanings and lived experiences surrounding participation in an EAI program. Beyond the perceived effects of EAI, our findings show that interactions with the horse, the setting of the sessions, and the way the intervention is organized play an equally important role. The results highlight the value of qualitative approaches in understanding how individuals experience and interpret their participation. Such methods offer a deeper insight into their experiences and place their voices back at the center of research.

For future research, it would be relevant to explore, from a qualitative perspective, how the contexts of interventions, whether carried out in external equestrian facilities or directly within institutions, shape the participants’ experiences. Such an approach would provide a better understanding of how the environment, the relationship with the horse, and institutional dynamics influence the meaning attributed to the intervention. It also remains essential to continue an approach that gives a voice to the beneficiaries of these programs to deepen our understanding of the experiential significance and subjective impact of these encounters for themselves and those around them.

## Figures and Tables

**Table 1 geriatrics-10-00145-t001:** Socio-demographic characteristics of the participants.

Variables	OA (*n* = 10)	FC (*n* = 10)	PC (*n* = 10)
Age	85.7 (4.83)	65.4 (9.17)	37.8 (12.1)
Sex	5F, 5M	5F, 5M	10F
Education level *	3.7 (1.16)	5.3 (1.73)	4 (0.89)
MMSE	14 (3.43)		

Notes: F = female; FC = family caregivers; M = male; MMSE = mini mental state examination; *n* = number; OA = older adults; PC = professional caregivers. * Education level was calculated by a nomenclature of diplomas by level: the International Standard Classification of Education (ISCED), which allows standardized reporting of education statistics according to an internationally agreed set of definitions and concepts, ISCED 1: Primary education, ISCED 2: Lower secondary education, ISCED 3: Upper secondary education, ISCED 4: Post-secondary non-tertiary education, ISCED 5: Short-cycle tertiary education, ISCED 6: Bachelor’s or equivalent level, ISCED 7: Master’s or equivalent level, ISCED 8: Doctoral or equivalent level.

**Table 2 geriatrics-10-00145-t002:** Thematic variance.

	Total, *n* = 30 (%)	OA, *n* = 10 (%)	FC, *n* = 10 (%)	PC, *n* = 10 (%)
Theme 1: Effects of EAI	100	100	100	90
Social interactions	80	70	100	70
Memory stimulation	70	90	90	30
Emergence of emotion	90	80	100	90
Behavior	53	80	30	50
Theme 2: Implementation of EAI	83	70	100	80
Institutional logistics	53	20	70	70
Leave the nursing home	43	10	50	70
Adherence	77	70	90	70
Theme 3: Environment of EAI	80	50	100	90
Break from daily routine	67	20	80	100
Contact with nature	27	20	20	40
Activity framework	23	30	30	10
Theme 4: Experience with horse	77	80	80	70
Relationship with the horse	67	80	60	60
Activity with the horse	53	80	50	30
Horse characteristics	40	60	30	30

Notes: EAI = Equine Assisted Intervention; FC = family caregivers; *n* = number; OA = older adults; PC = professional caregivers.

## Data Availability

Data available on request from the authors: The data that support the findings of this study are available from the corresponding author upon reasonable request.

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
