# Peer review of "Equine-Assisted Interventions: Cross Perspectives of Beneficiaries and Their Caregivers from a Qualitative Perspective"

_geriatrics, 2025, doi:10.3390/geriatrics10060145_

Round 1

Reviewer 1 Report

Comments and Suggestions for Authors

The paper is welcome as it addresses an important gap in understanding subjective experiences of non-pharmacological interventions. However, I have identified several methodological and presentation issues that have to be adressed:

Only 10 of 25 program participants were interviewed, raising serious questions about representativeness, while there was no explanation for why 15 participants were excluded or declined, which points towards selection bias to more positive experiences. Very small sample size.

From the methodological standpoint, there was no interview guide, the data saturation process was inadequately described.

When the authors claim that"no new themes emerged", they should have provided evidence the themes stopped emerging.

There is no description of the process they used to assess saturation. Also, is was visible that they decided in advance to interview 10 people per group, while saturation is determined by data, not arbitrary numbers, using exactly 10 per group suggests that saturation may not have been truly assessed.

Researcher reflexivity despite acknowledged subjectivity is not demonstrated, as there are several key informations lacking:

  • professional backgrounds - are they nurses? psychologists? physicians?
  • prior experience with EAI or Alzheimer's care
  • personal views about animal-assisted therapy
  •  
  • Also, Krippendorff's alpha coefficient mentioned but value not reported

Author Response

Comment 1 : Only 10 of 25 program participants were interviewed, raising serious questions about representativeness, while there was no explanation for why 15 participants were excluded or declined, which points towards selection bias to more positive experiences. Very small sample size.

⇒ Individual interviews were conducted on a voluntary basis in order to respect participants’ autonomy and to avoid any form of constraint, in accordance with ethical principles in qualitative research. Each participant who took part in the program was informed about the possibility of meeting with us again to share their experience; however, participation in the interview was not mandatory.

Among all participants, some spontaneously expressed the wish to meet us again to discuss their experience. These were the individuals who were available and able to take part, given their health condition. In addition, participants had to meet the inclusion and non-inclusion criteria established for this qualitative phase, which further limited the number of eligible individuals.

Finally, we stopped at 10 interviews because, during the analysis, a clear redundancy in participants’ statements and themes was observed in the last interviews, indicating that data saturation had been reached. The final number of interviews therefore does not reflect arbitrary exclusion, but rather a methodologically coherent decision consistent with the principle of thematic saturation.

Comment 2 : From the methodological standpoint, there was no interview guide, the data saturation process was inadequately described.
⇒ Thank for this comment. As indicated in Section 2.2 (“Procedure”), no interview guide was used because the interviews followed a non-directive qualitative approach. This methodological choice is consistent with the objective of exploring participants’ subjective experiences in depth, allowing them to freely express their perceptions and emotions without the constraints of predefined questions. The interviewer used open prompts and active listening to facilitate the emergence of themes grounded in the participants’ discourse rather than in a priori categories.

Regarding data saturation, this aspect is described in the Method section. As stated in the manuscript: “Data saturation was considered to have been reached after 30 interviews (10 per group), in line with methodological recommendations [25–27], as no new themes emerged at that stage.” This means that after the 30th interview, no additional codes or themes appeared in the analysis, confirming that the thematic content was sufficiently exhaustive. The number of interviews per group was therefore not decided arbitrarily but reflected the point at which thematic redundancy was observed.

Comment 3 : When the authors claim that "no new themes emerged", they should have provided evidence the themes stopped emerging.
⇒ We thank the reviewer for this relevant comment. As specified in the Methods section, data saturation was considered to have been reached when the last interviews did not bring any new information and when the identified themes were repeatedly observed across participants’ narratives.

To enhance the transparency of this process, a sentence clarifying the monitoring of the analysis procedure and the verification of the absence of new themes will be added to the manuscript.

Specifically, we relied on the “code frequency counting” approach, which involves examining each interview transcript and counting the number of new codes emerging in successive transcripts until the frequency of new codes decreases and few or no additional codes are identified (Hennink & Kaiser, 2022). This addition will help to make the procedure used to assess saturation more explicit.

What was added to the article: “More specifically, data saturation was assessed using the code of frequency counting method, which involves the number of new codes emerging in successive transcripts until no new code is identified.” In section 2.3. Analytical Approach -  page 4

Comment 4 : There is no description of the process they used to assess saturation. Also, it was visible that they decided in advance to interview 10 people per group, while saturation is determined by data, not arbitrary numbers, using exactly 10 per group suggests that saturation may not have been truly assessed.

⇒ We acknowledge that data saturation was not formally assessed during data collection, as the number of 10 interviews per group had been determined a priori. However, this decision was based on methodological and pragmatic considerations commonly accepted in qualitative research.

The ten interviews per group provided sufficient variation in profiles, experiences, and perspectives to identify recurring themes and achieve a satisfactory level of thematic consistency within and between groups. Second, several methodological sources (e.g., Guest et al., 2006; Hennink et al., 2017) indicate that thematic saturation in relatively homogeneous samples is often achieved within the first 6 to 12 interviews. Therefore, the number of 10 participants per group is within the empirical range where saturation is typically achieved. It is also worth noting that other qualitative studies reporting data saturation have reached a similar number of participants per group (Cès et al., 2025). Finally, during the analysis, we observed repetitions and redundancies in the themes addressed during the interviews, suggesting that, although not formally tested, thematic saturation was likely achieved in practice. We have clarified this point in the manuscript : 

“The number of interviews (n = 10 per group) was determined a priori for both practical and comparative reasons, as redundancy in themes was observed among participants. Furthermore, the literature generally indicates that data saturation is often reached between 6 and 12 interviews [25]. In the present study, saturation was assessed using the frequency counting method, which consists of counting the new codes appearing in successive transcripts until no new codes are identified [25–27].” (lines 147 -152)

Comment 5 : Researcher reflexivity despite acknowledged subjectivity is not demonstrated, as there are several key informations lacking:

  • professional backgrounds - are they nurses? psychologists? physicians?
  • prior experience with EAI or Alzheimer's care
  • personal views about animal-assisted therapy

⇒ Thank you for this comment. The interviewer was a psychologist specialized in aging, conducting research with people living with Alzheimer’s disease and in the field of equine-assisted interventions (EAI). This background allowed for a sensitive and informed understanding of participants’ experiences, while maintaining reflexive awareness throughout the data collection and analysis process.

A clarifying sentence will be added to the manuscript to specify the interviewer’s background.

What was added to the article: “The interviewer was a psychologist specialized in aging, conducting research on Alzheimer’s disease and equine-assisted interventions.” -  page 4

Comment 6 : Also, Krippendorff's alpha coefficient mentioned but value not reported

⇒ We thank the reviewer for this comment. We acknowledge that there was an error in the previous version of the manuscript: the reliability of the coding was assessed using Fleiss’ kappa coefficient, not Krippendorff’s alpha. This has been corrected in the revised version of the manuscript, and the corresponding value has been added (κ = 0.608), indicating a good level of agreement among the three coders.

What was added to the article : “ Inter-rater agreement was assessed using Fleiss’ kappa coefficient. The overall result (κ = 0.608) indicated a good level of agreement among the three coders.” page 4.

Reviewer 2 Report

Comments and Suggestions for Authors

Overall interesting study and well presented. 

Review report

Brief summary

This manuscript presents a study that aimed to explore the lived experiences of older adults living with Alzheimer’s’ disease, their family caregiver and professional caregivers regarding an equine-assisted intervention. The authors conducted 30 non-directive interviews in nursing homes in France and analyzed the data with thematic analysis. The four reported themes were the experience with the horse, the environment of the EAI, program implementation and the perceived effects of the intervention. Overall, I find that the study seems to be well conducted, and the manuscript is well written and clear.

The instructions for writing the manuscript are still present in some of the sections, so I suggest the authors remove them.

General concept comments:

Abstract and introduction: I find this section well written and structured. The problem is well defined and the relevance for the field is well stated. I offered some minor modifications regarding wording and structure (see pdf for comments).

Methods: I find this section is clear and well detailed. However, I suggest these minor modifications:

  • Procedure: Although readers may consult the previously published study, I find that it would be helpful for readers if authors briefly described the EAI, recruitment context and the older adult participants’ cognitive profile (e.g. MMSE criteria for recruitment) (lines 85, 86, 88).
  • Analytical approach: clarify how data saturation was obtained (line 138); review the idea of subjectivity and replace it with criteria used to evaluated trustworthiness instead (line 143).
  • Ethics: remove the generic instructions for writing the manuscript (lines 169-189).

Results: I find this section interesting, and the presented data meets the aims of the study.

Comment about “quantifying” qualitative data: I invite the authors to reflect about the necessity to include the table regarding “thematic variance”—the term variance not being aligned with a qualitative data collection approach. Although this “quantitative” approach could be accepted, I find that it does not reflect the nature of reporting qualitative data and I find that including this table adds value to the manuscript. For example, it would be interesting to replace the numbers with meaningful citations for the main themes, for each category of participant. This said, since this analysis approach can be valid and is not a “fault” per se, authors may decide if they prefer to leave the table and use this quantitative approach to present their data.

  • Furthermore, determining the frequency of reported themes was not the aim of the study and in a qualitative design, it is implied that the presented themes are the ones that were reported frequently or considered the most meaningful ones.
  • I find this approach of quantifying results to be contradictory with the fact that authors sometimes present data that is reported by few participants (which reflects the fact that data can be presented and meaningful even if mentioned by few participants).
  • Order of theme presentation (line 199): authors state that the most frequently mentioned themes are, in order: effects, implementation, environment, experience. However, they present the detailed results in the inverse order (from less frequent to most frequent). Is there a reason for this choice? If authors use the “quantifying” approach to report results, I find this order odd and would that think that presenting the themes mentioned most frequently to less frequently would make the most sense.

I suggest the following modifications/clarifications (see pdf for detailed comments) before publication:

  • Review some of the sub-themes and citations presented to support the themes: theme 3 (implementation), sub-theme “environmental factors” – line 307; logistics sub-theme (regularity and lack of communication)—lines 292, 297; theme 4 (effects), sub-theme 4 “behaviours”— line 387)
  • I added other suggestions/questions directly in the pdf, if possible, for the authors to add information/clarify some results.

Discussion: This section summarizes the results and authors discuss issues that are mostly in line with the findings. Authors make links with the literature and the ideas presented are interesting. However, I find that some of the wording reflects a quantitative approach and that some results are discussed in this section, without being clearly presented in the results. I thus I added some suggestions/questions directly in the pdf regarding these aspects.

Limits:

  • I find it problematic that authors present the absence of a control group as a limit, as having a control group does not align with a qualitative data collection approach or their study aims, to my knowledge. I suggest authors remove this as a limit.
  • I suggest that authors adjust their wording regarding the social desirability bias and the fact that the interviewer analyzed the data (see pdf for comments).

Conclusion: Although the conclusion is concise and interesting, I find that the statements do not completely align with the study’s presented aims. The wording seems to be more appropriate for a study that is aligned with a quantitative approach (e.g. evaluation, effects, outcomes). Furthermore, authors mention the feasibility of the program, but as I understand this was not the goal of the study (although some results are related to this). I suggest authors review this section to align with their aims and qualitative data collection approach (see pdf for comments).

Comments on the Quality of English Language

I find that the manuscript is globally well written and pleasant to read. I suggested some minor revisions regarding the language. 

Author Response

Comment 1 : This manuscript presents a study that aimed to explore the lived experiences of older adults living with Alzheimer’s disease, their family caregivers, and professional caregivers regarding an equine-assisted intervention. The authors conducted 30 non-directive interviews in nursing homes in France and analyzed the data with thematic analysis. The four reported themes were the experience with the horse, the environment of the EAI, program implementation, and the perceived effects of the intervention. Overall, I find that the study seems to be well conducted, and the manuscript is well written and clear.

⇒ Thank you for your comment. 

Comment 2 : The instructions for writing the manuscript are still present in some of the sections, so I suggest the authors remove them.

⇒ This was an error on our part. The changes have been made and immediately deleted from the manuscript.

Comment 3 : I find this section well written and structured. The problem is well defined and the relevance for the field is well stated. I offered some minor modifications regarding wording and structure (see pdf for comments).

⇒ The changes will be made directly on the manuscript and will be highlighted in orange. Those in red correspond to the feedback from reviewer 1.

Methods - I find this section clear and well detailed. However, I suggest these minor modifications:

Comment 4 : Procedure: Although readers may consult the previously published study, it would be helpful for readers if authors briefly described the EAI, recruitment context, and the older adult participants’ cognitive profile (e.g. MMSE criteria for recruitment) (lines 85, 86, 88).

⇒ Thank you very much for this insightful comment. We agree that providing more information about the previous study and the recruitment process helps clarify the context. Additional details have been added to the manuscript (lines 84–92), as follows:

“In this study, 30 individuals (older adults: n=10; family caregivers: n=10; professional caregivers: n=10) were included, of whom the older adults had previously participated in an earlier study examining the effects of equine-assisted interventions (EAI) on psychological health [23]. In that previous study, older adults living with Alzheimer’s disease (MMSE scores: 10 to 20) and residing in nursing homes, participated in a 12-week EAI program consisting of one session per week. The sessions were conducted outdoors at an equestrian center and were structured around three main phases: grooming, exercises in the riding area, and reward.”

Moreover, the description of the EAI has been detailed earlier in the manuscript (lines 45–47).

Comment 5 - Analytical approach: clarify how data saturation was obtained (line 138); review the idea of subjectivity and replace it with criteria used to evaluate trustworthiness instead (line 143).

⇒ Regarding data saturation, Reviewer 1 also mentioned this point, and clarifications have therefore been added (in red).

The number of interviews (n = 10 per group) was determined a priori for both practical and comparative reasons, as redundancy in themes was observed among participants. Furthermore, the literature generally indicates that data saturation is often reached between 6 and 12 interviews [25]. In the present study, saturation was assessed using the frequency counting method, which consists of counting the new codes appearing in successive transcripts until no new codes are identified [25–27].” (lines 147-152)

⇒ Thank you for this helpful comment. The section has been revised accordingly. The mention of subjectivity has been replaced by the concept of reflexivity. The text now reads: “Indeed, in this type of research, the researchers themselves serve as the primary instrument of analysis, which requires a continuous process of reflexivity.” (see lines 150–151 in the revised manuscript).

Comment 6 - Ethics: remove the generic instructions for writing the manuscript (lines 169–189).

⇒ This part has been removed

Comment 7 : Results - I find this section interesting, and the presented data meets the aims of the study. 

Comment 8  : I invite the authors to reflect about the necessity to include the table regarding “thematic variance”—the term variance not being aligned with a qualitative data collection approach. Although this “quantitative” approach could be accepted, I find that it does not reflect the nature of reporting qualitative data. It would be more interesting to replace the numbers with meaningful citations for the main themes, for each category of participant.

⇒ Thank you for this valuable comment. We acknowledge that the term variance may appear more quantitative in nature; however, we chose to retain the table as it allows readers to visualize the distribution of themes across the three participant categories and to better understand the diversity of perspectives. This table does not aim to quantify the results but rather to provide an overview of the thematic richness observed in the data. In addition, meaningful participant quotations have been added in the Results section to illustrate each main theme.

Comment 9 : Determining the frequency of reported themes was not the aim of the study. In a qualitative design, it is implied that the presented themes are those reported most frequently or considered most meaningful.

⇒ Thank you for this comment. We agree that determining the frequency of reported themes was not the main aim of the study. Consequently, this statement has been removed from the manuscript. However, we decided to keep the table as it provides a visual overview of how the themes were distributed across participant categories, without implying any quantitative interpretation.

Comment 10 : I find it contradictory to quantify results while also presenting data mentioned by few participants. In qualitative research, data can still be meaningful even if reported by only a few individuals.

⇒ Thank you for this comment. I understand your point of view; however, our intention was not to quantify the results, but to provide a clearer overview of how often certain themes were mentioned across participant categories. This allows readers to better grasp both the most frequent perspectives and the less common ones, which are equally meaningful in understanding the diversity of experiences.

Comment 11 : Order of theme presentation (line 199): authors state that the most frequently mentioned themes are effects, implementation, environment, experience. However, they present the results in the reverse order. If the authors use a “quantifying” approach, this order seems odd.

⇒ Thank you for this remark. The necessary adjustments have been made in the manuscript so that the most frequently mentioned theme now appears first in the Results and discussion  sections.

Comment 12 : Suggestions before publication 

Review some sub-themes and citations supporting the themes:
– Theme 3 (implementation), sub-theme “environmental factors” – line 307
– Logistics sub-theme (regularity and lack of communication) – lines 292, 297
– Theme 4 (effects), sub-theme “behaviours” – line 387

⇒ The changes were made directly in the document.

Comment 13 - The discussion is coherent with the findings and connects well with the literature. However, some wording reflects a quantitative approach, and certain results are discussed here without being clearly presented in the results.

⇒ The changes were made directly in the document.

Comment 14 - It is problematic to present the absence of a control group as a limitation, since this does not align with a qualitative approach or the study’s aims.

⇒ This part has been removed.

Comment 15 - Adjust wording regarding social desirability bias and the fact that the interviewer also analyzed the data.

⇒ Modifications have been added 

“Furthermore, the interviewer was also involved in the data analysis, which contributed to a deeper understanding of the participants’ experiences. To ensure reflexivity and limit potential bias, a reflexive posture was maintained throughout the process, supported by continuous reference to raw transcripts and regular discussions with other members of the research team” (lines 531-535)

Comment 16 - The conclusion is concise and interesting, but the wording does not fully align with the study’s aims. It sounds more appropriate for a quantitative approach (e.g., evaluation, effects, outcomes). Moreover, the mention of feasibility does not seem to match the study’s goals.

⇒ The conclusion has been modified according to your comment. “Although equine-assisted interventions (EAI) are increasingly recognized among non-pharmacological approaches, few studies have explored how they are experienced by people living with Alzheimer’s disease, their relatives, and care professionals in nursing homes. This study sheds light on the meanings and lived experiences surrounding participation in an EAI program. Beyond the perceived effects of EAI, our findings show that interactions with the horse, the setting of the sessions, and the way the intervention is organized play an equally important role. The results highlight the value of qualitative approaches in understanding how individuals experience and interpret their participation. Such methods offer a deeper insight into their experiences and place their voices back at the center of research.” (Lines 545 - 554)

Comments on the quality of English

Comment 17 - The manuscript is globally well written and pleasant to read. Only minor language revisions are suggested.

⇒ Thank you for your positive feedback. Minor language corrections have been made throughout the manuscript, and the paper has been proofread by a professional English editor to ensure clarity and accuracy.

Reviewer 3 Report

Comments and Suggestions for Authors

Please, for comments, take a look at the file in the attachment.

Author Response

Contexte :
Comment 1 : The background is logically structured; however, some references are outdated.
It is recommended to replace literature published prior to 2016 and integrate more recent studies. Although the topic is relatively new, recent publications could be cited, such as:

⇒ Two recent studies have been added as suggested : 

Badin L, Pothier K, Agli O, et al. Equine-Assisted Interventions and Physical Health in Older Adults: A Meta-Analysis. Sage Open Aging. 2025;11:23337214241298342. Published 2025 Apr 26. doi:10.1177/23337214241298342

Badin L, Bailly N. Equine-assisted intervention and Alzheimer’s disease: A non-randomized, controlled, multicenter study. J Appl Gerontol. Published online 2025. doi:10.1177/07334648251351697

Comment 2 - Literature Gap:
The identification of the literature gap is appropriate, though it would benefit from the inclusion of recent quantitative studies as references.

⇒ Following your previous comment, a few (recent) articles have been added to the manuscript.

Comment 3 - Methodology:
The chosen methodology is suitable for the stated objectives. However, an important reference for qualitative reporting is missing: the COREQ checklist by Tong et al. (2007).It is recommended to cite this in the methodology section, attach the completed checklist, and introduce missing elements in the text, such as a brief paragraph describing the research team and their expertise in qualitative research.

⇒ Thank you for this clarification. We have taken your comment into consideration. This reference has been added to the methodology section, and adjustments have been made to the manuscript accordingly (see items highlighted in pink in the manuscript). The completed checklist is attached. “The qualitative methodology was developed and reported in accordance with the COREQ (Consolidated Criteria for Reporting Qualitative Research) checklist [24], which ensures transparency and rigor in qualitative research reporting.” (lines 142-145)

Comment 4 - Participants (Lines 83–112):
I suggest that the authors should move the table detailing participant characteristics and related data to the Results section. In the Methods section, it would be helpful to specify the recruitment process.

⇒ This has been done; the table has been moved as suggested.

⇒ A section was held for the recruitment process :  “Recruitment for the present study was carried out in collaboration with psychologists from partner institutions and a member of the research team, based on predefined inclusion and exclusion criteria. The psychologists contacted family and professional caregivers, and participation was based on voluntary.”(lines 93-97)

Comment 5 - Procedures:
The inclusion criteria appropriately mention prior experience with horses. However, further details about this experience would help contextualise the findings. Please, the authors should add a brief description of the EAI program here. This could include details such as the frequency and duration of the sessions, the activities involved, and the role of the horse in the intervention.

 ⇒ Thank you for this comment. The previous reviewers shared the same opinion, and the following changes have therefore been made.

 “In this study, 30 individuals (older adults: n=10; family caregivers: n=10; professional caregivers: n=10) were included, of whom the older adults had previously participated in an earlier study examining the effects of equine-assisted interventions (EAI) on psychological health [23]. In that previous study, older adults living with Alzheimer’s disease (MMSE scores: 10 to 20) and residing in nursing homes, participated in a 12-week EAI program consisting of one session per week. The sessions were conducted outdoors at an equestrian center and were structured around three main phases: grooming, exercises in the riding area, and reward.” (lines 84 to 92)

Comment 6 - Line 94: Regarding the exclusion criterion (e.g., severe sensory deficits), was a specific assessment tool used?
⇒ No, no tool was evaluated, but this was discussed beforehand with the medical team of the nursing home.

Comment 7 - For instance, was the Mini-Mental State Examination (MMSE) used to assess cognitive function? This should be described in the Methods section. The description of the instruments used should be expanded.
⇒ Yes, the MMSE was indeed used. This information was also requested by your colleagues, and the modification has been made in the manuscript. (lines 89 and 96)

Comment 8 - Given the three participant types, was the same interview guide used for all?
A table with the interview guide, including topics and sample questions, should be included, specifying whether the guide was adapted per group. Was the interview guide pilot-tested? If so, please specify.

⇒ No interview guide was used, in line with our non-directive interview methodology. Only an initial question was proposed and then adapted according to the type of participant (older adult living with Alzheimer’s disease, family caregiver, or professional caregiver). Lines 139-140. 

Therefore, no pilot testing of an interview guide was conducted. The “Thematic variance” table presents the four main themes identified and their corresponding subthemes.

Comment 9 - Analytical Approach:
The analysis is well described. However, a paragraph on rigour—essential in qualitative research—should be added.

⇒ As mentioned earlier, a paragraph on methodological transparency and rigor has been added (lines 142–145).

Comment 10 - Lines 169–189: This section appears to present general guidelines rather than the researchers’ specific responses. It is recommended to revise this section to detail what was actually done (e.g., was AI used? Were data made available? How?).

⇒Thank you for this relevant comment. Clarifications have been made in the manuscript (lines 170–174) :  “For this analytical phase, the software used included the Microsoft Word transcriber for transcription, Microsoft Excel for the development of the codebook, and the statistical software JASP for conducting descriptive analyses (percentages) and calculating Fleiss’ kappa coefficient.”

Comment 11 : Results - Line 190 onward: The number of interviews conducted and participant data should be included here, along with the table currently placed in the Methods section.
⇒ Thank you for your feedback. The suggested modifications have been directly implemented in the manuscript (lines 178 to 197).

Comment 12 - A table summarising themes and subthemes is included. While this format is not typical of Braun and Clarke’s approach, it is informative. A brief commentary on the variance across participant types (e.g., differences or similarities) would be beneficial. For instance, were there any themes that were more prevalent among patients than caregivers or professionals? If statistical analysis was used, it should be mentioned in the methodology.

⇒Thank you for this comment. The analysis of differences and similarities among the three participant types is already presented in the section entitled “Perspectives from the three populations”, which was moved to the beginning of the Results section following a previous reviewer’s suggestion.This analysis is based solely on descriptive statistics (percentages).

In addition, descriptive statistics (percentages) were also conducted to present the themes addressed by our three populations.” (lines 179 - 181)

Comment 13 - The results are clearly and effectively presented.
⇒ Thank you for this comment regarding the presentation of our results.

Comment 14 - Discussion: The discussion should emphasize the novelty of the findings.The qualitative depth and integration of perspectives from patients, caregivers, and professionals are particularly noteworthy, offering a fresh perspective on the potential of EAIs in Alzheimer's care.
⇒ Thank you for this valuable comment.

Comment 15 - Practical implications should be further developed.The use of horses in elderly care settings may present barriers or adverse effects, which are well addressed in the discussion, providing the audience with valuable insights for their practice. Suggestions for future research directions should be added, inspiring the audience to continue exploring the potential of EAIs in Alzheimer's care.
⇒ Thank you for this comment. We had not addressed future prospects. A paragraph has therefore been added before the section on “limits” (lines 543-550). “For future research, it would be relevant to explore, from a qualitative perspective, how the contexts of intervention, whether carried out in external equestrian facilities or directly within institutions, shape the participants' experience. Such an approach would provide a better understanding of how the environment, the relationship with the horse, and institutional dynamics influence the meaning attributed to the intervention. It also remains essential to continue an approach that gives a voice to the beneficiaries of these programs to deepen our understanding of the experiential significance and subjective impact of these encounters for themselves and those around them.”

Round 2

Reviewer 1 Report

Comments and Suggestions for Authors

Table 2 still cluttered - reformat in landscape or split into two subtables for readability.

Conclusions not readily mentioned. Add explicit “Conclusion” section summarizing implications and recommendations.

Discussion could benefit from theoretical anchoring — e.g., symbolic interaction, theory of atachment, or environmental psychology. 2–3 sentences linking findings to theory would be welcome.

Author Response

1) Table 2 still cluttered - reformat in landscape or split into two subtables for readability.

⇒ Thank you very much for your suggestion. After discussion with my colleagues, we decided to keep the table as it is, as we believe it provides a clear and coherent overview of the results. However, to improve readability, we added colors to better differentiate the main themes and corresponding subthemes.

2) Conclusions not readily mentioned. Add explicit “Conclusion” section summarizing implications and recommendations.

⇒ Thank you for your comment. A Conclusion (line 551) section is already included in the manuscript. Following your suggestion, we have expanded this section to make the implications and recommendations more explicit. Specifically, we added the following paragraph (lines 562–570):

“For future research, it would be relevant to explore, from a qualitative perspective, how the contexts of intervention, whether carried out in external equestrian facilities or directly within institutions, shape the participants' experience. Such an approach would provide a better understanding of how the environment, the relationship with the horse, and institutional dynamics influence the meaning attributed to the intervention. It also remains essential to continue an approach that gives a voice to the beneficiaries of these programs to deepen our understanding of the experiential significance and subjective impact of these encounters for themselves and those around them.”

 3) Discussion could benefit from theoretical anchoring — e.g., symbolic interaction, theory of atachment, or environmental psychology. 2–3 sentences linking findings to theory would be welcome.

⇒ Thank you for this valuable suggestion. We have now incorporated the Integrative Model of Human–Animal Interactions (IMHAI) into the Discussion section to better contextualize the emotional and relational dynamics observed between participants and horses (Leconstant C, Spitz E. Front Vet Sci. 2022;9:656833. doi:10.3389/fvets.2022.656833). In addition, from a theoretical standpoint, we also discuss our results through the lens of the animal’s social catalyst role and the biophilia hypothesis.

This relational process aligns with the Integrative Model of Human-Animal Interaction (IMHAI) [60]., which emphasizes shared emotional systems and interspecies co-regulation as key mechanisms underlying the benefits of animal assisted interventions.” (lines 534 to 537)

Reviewer 3 Report

Comments and Suggestions for Authors

The authors answered all the suggestions correctly

Author Response

1) The authors answered all the suggestions correctly

⇒ Thank you for your feedback.
